# Functionality and Usability of mHealth Apps in Patients with Peritoneal Dialysis: A Systematic Review

**DOI:** 10.3390/healthcare12050593

**Published:** 2024-03-05

**Authors:** Shu-Mei Chao, Chao-Kuei Pan, Ming-Ling Wang, Yu-Wen Fang, Shu-Fen Chen

**Affiliations:** 1Department of Nursing, Tzu Chi University of Science and Technology, Hualien 970302, Taiwan; shumei@ems.tcust.edu.tw (S.-M.C.); ss107@ems.tcust.edu.tw (C.-K.P.); 2School of Nursing, International Ph. D Program in Gerontology and Long-Term Care, Taipei Medical University, Taipei 110301, Taiwan; aileen22@tmu.edu.tw; 3Department of Health Administration, Tzu Chi University of Science and Technology, Hualien 970302, Taiwan; 4Department of Nursing, Shuang Ho Hospital, Taipei Medical University, New Taipei City 235041, Taiwan

**Keywords:** mHealth, peritoneal dialysis, functionality, usability

## Abstract

mHealth has been utilized in the care of patients with chronic kidney disease, allowing the collection of patient health-related data, offering disease-related information, enabling the tracking and recording of biochemical parameters, and enabling communication with healthcare providers in real time through applications. mHealth may improve the health outcomes in patients with peritoneal dialysis. This systematic review aimed to summarize evidence regarding the functionality and usability of mHealth apps in patients with peritoneal dialysis. We conducted a comprehensive literature review, searching in five databases, including CINAHL, Cochrane, PsycINFO, PubMed, and Web of Science, to retrieve titles and abstracts related to peritoneal dialysis and mHealth applications for PRISMA recommendations from January 2013 to December 2023. Overall, 11 studies met all the inclusion criteria. The functionality of mHealth apps included inform, instruct, record, display, guide, remind/alert, and communicate. Most of the apps have multifunctionality. The usability was categorized into three aspects: efficiency (self-efficacy and usability), satisfaction, and effectiveness (underwent kidney transplantation and switched to hemodialysis, rehospitalization, peritonitis rate, infection rates at exit sites, mortality, fluid overload, inadequate solute clearance, biochemical values, quality of life, consumer quality index, and technology readiness). Generally, outcomes in the intervention group had better effects compared to those in the control group. Multifunctional mHealth apps show a good potential in improving the efficiency, satisfaction, and effectiveness for patients compared to traditional care. Future research should include more studies and participants to explore and verify the long-term effectiveness of mHealth apps.

## 1. Introduction

Unlike hemodialysis (HD), peritoneal dialysis (PD) does not require patients to visit hospitals for dialysis two to three times a week, allowing them to maintain their regular work and social activities, preserving their independence and freedom [1]. Additionally, the cost of hemodialysis was 1.37 times compared to peritoneal dialysis in Taiwan, and opting for PD can reduce healthcare costs. Despite these advantages, most patients continue to prefer HD [2]. The reluctance toward PD may stem from the fact that it involves self-administration or caregiver assistance, requiring patients to manage the dialysis process, including attaching the dialysis bag and manually recording related data on paper [3], which can be challenging for some patients.

While patients can communicate with healthcare providers (HCP) over the phone to address any issues during daytime dialysis [4], seeking advice from them at night may pose problems due to their unavailability. Furthermore, for patients without a medical background, HCP must impart additional disease-related knowledge and provide opportunities for them to practice dialysis techniques. The lack of a medical background and sudden complications can cause patients to feel overwhelmed, resulting in them choosing HD over PD [1]. Mobile health (mHealth) applications (apps) have the potential to increase patients’ willingness to adopt PD. These apps can record PD conditions, enhance patients’ self-practice of PD techniques at home, alleviate concerns about unmanageable health conditions, help patients with chronic kidney diseases understand their condition, strengthen disease self-management [5,6,7], facilitate two-way communication and data sharing between patients and HCP, assist patients in gaining necessary knowledge and skills, and support patients in problem solving and decision making [1,8,9,10,11], ultimately increasing patients’ sense of security and satisfaction.

mHealth provides healthcare services through portable devices such as smartphones, tablets, computers, and personal digital assistants [12,13]. mHealth delivers pre-established health services and provides guidance regarding the PD procedure and disease-related information in real time through features such as browsing, text messaging, communication, and videos [11,14,15]. mHealth apps enable the collection of patient health-related data, including access to personal medical records, obtaining nutritional or dietary information, receiving treatment-related details, offering disease-related information, understanding health status, tracking exercise activities, weight, and dietary habits, monitoring medication dosage, recording blood pressure, and providing convenience and efficiency for patients with PD [3,4,6,9,14,16,17,18,19]. Additionally, some mHealth apps have interactive features that allow patients to communicate immediately with physicians or nurses when difficulties arise [6]. Upon identifying problems in patient data transmissions, HCP can proactively contact the patients. These features contribute to patients feeling reassured and maintaining their independence [3].

However, concerns regarding issues such as unsuitable design, data reliability [13,20], inadequate user technological literacy, and privacy and security [3,19] may affect patients’ willingness to use mHealth services, and the efficacy of such services remains unclear. If mHealth apps lack a scientific basis, certification, or regulatory requirements, they may be unsuitable or potentially harmful for patients [8]. A systematic literature review of the effectiveness of eHealth for patients with PD showed inconsistent results regarding infection rates, hospitalization outcomes, knowledge, skills, satisfaction, and quality of life [8]. However, van Eck van der Sluijs et al. (2021) suggested that eHealth can reduce the incidence of peritonitis [16]. Therefore, to determine the overall impact of mHealth apps on patients with PD, a comprehensive analysis through systematic evaluation is necessary. Thus, this study aimed to conduct a systematic review to explore the features, functions, and usability of mHealth apps used by patients with PD.

In terms of usage, each mobile app emphasizes different things, making it challenging to establish a standard for systematic evaluation [13]. To clarify the scope of assistance provided by mHealth apps for patients with PD, this study focused on the usability and functionality of mobile apps. Regarding functionality, we referred to the seven functions proposed by the Institute for Healthcare Informatics (IMS) for mHealth apps: inform, instruct, record, display, guide, remind/alert, and communicate [21]. These functions have been widely evaluated in mHealth apps, such as those for asthma [22], chronic kidney diseases [17], and heart failure [23]. In terms of usability assessment, Harrison et al. (2013) [24] and Weichbroth (2020) [25] advocated that an ideal mHealth app should be simple and user-friendly. According to Weichbroth (2020), an app’s usability can be assessed by exploring three aspects: efficiency, satisfaction, and effectiveness [25]. Therefore, in this study, we explored the aforementioned aspects in order to evaluate and analyze the usability of mHealth apps.

mHealth apps have been widely used to enhance the care of patients with PD. However, whether they effectively improve health requires a comprehensive and concrete analysis. Therefore, this systematic review aimed to investigate studies on the use of mHealth apps in patients with PD regarding the following aspects: (1) characteristics, (2) functionalities, and (3) usability.

## 2. Materials and Methods

This systematic review was reported in accordance with PRISMA guidelines, and was conducted by two reviewers following the systematic review and meta-analysis expansion methodology [26,27]. This review aimed to describe literature retrieval and study selection using the PRISMA checklist for reporting systematic reviews. In cases of inconsistencies in the results, a third reviewer was consulted. This review involved mapping the entirety of the relevant literature to identify and map features, functionalities, and usability related to mHealth apps for PD. The findings contribute not only to identifying empirical patterns and resources for clinical practice but also to recognizing the existing limitations of apps for future enhancement.

### 2.1. Search Strategy

Between January 2013 and December 2023, five databases—CINAHL, Cochrane, PsycINFO, PubMed, and Web of Science—were searched to retrieve titles and abstracts related to PD and mHealth apps. The search strategy incorporated search strings such as “mHealth”, “mobile health”, “mobile app”, “mobile application”, “smartphone application”, “app”, “apps”, “telemedicine”, “peritoneal”, “dialysis”, and “peritoneal dialysis” by Boolean operators. The following criteria were applied for selecting research articles: (1) full text; (2) English language; (3) studies classified according to Joanna Briggs Institute (JBI) levels of evidence, including experimental designs, quasi-experimental designs, observational–analytic designs, observational–descriptive studies, expert opinion, and bench research [28]; and (4) the exclusion of study protocols, scoping literature, literature reviews, systematic reviews, integrative analysis reviews, conference abstracts, letters to the editor, and editorials and similar articles.

### 2.2. Inclusion and Exclusion Criteria

A review of the use of mHealth apps in PD was conducted. The inclusion criteria were as follows: (1) apps targeted at patients with PD; (2) mobile app interventions delivered through smartphones, tablets, or web portals accessible via mobile devices; (3) apps aimed at enhancing patient care and the management of the disease. The exclusion criteria were as follows: (1) eHealth remote healthcare, electronic health records, and electronic medical records; and (2) designed and developed without adoption in patients.

### 2.3. Screening and Data Extraction

Title, abstract, and full-text screening was independently conducted by two reviewers (CS and FY). All identified articles were uploaded to EndNote 20, and duplicate articles were removed. The two reviewers used three predesigned forms to extract data from the eligible articles. The first form gathered the following information: author/year/country, research objectives, app name/device/platform/objectives, design/sample size/duration, intervention content, and outcome variables/results. The second form assessed the functionality of mHealth apps by utilizing the seven functions proposed by IMS: inform, instruct, record, display, guide, remind/alert, and communicate. The third form focused on evaluating the apps’ usability by exploring the aspects of efficiency, satisfaction, and effectiveness. The extracted data were synthesized and analyzed using the three aforementioned forms.

### 2.4. Selection Process

A total of 184 articles were extracted from the five databases. After removing duplicates (n = 10) as well as abstracts and articles that did not meet the inclusion criteria, 152 articles were excluded, leaving 22 articles for consideration. Two independent reviewers conducted the screening based on the inclusion and exclusion criteria and resolved any discrepancies through discussions. Based on the inclusion and exclusion criteria, 11 articles were subsequently excluded, resulting in a final set of 11 articles for the systematic review. The search results and study inclusion process are presented in the Preferred Reporting Items for Systematic Reviews and Meta-Analyses flowchart (Figure 1).

### 2.5. Functionality Assessment

The app’s functionality was evaluated using the guidelines developed by the IMS [21] as follows: (1) inform: provide information in the form of text, photos, videos, etc.; (2) instruct: offer instructional guidance; (3) record: gather physiological data and other relevant patient information; (4) display: show data entered by users and recipients in a graphical format; (5) guide: analyze user-inputted data and suggest relevant treatments; (6) remind/alert: send reminders or alerts to notify patients; (7) communicate: patients interact with HCP, peers, or community forums. These functionalities serve various purposes and enhance the overall capabilities of mHealth apps.

### 2.6. Usability Assessment

The usability of mHealth apps can be assessed by exploring their efficiency, satisfaction, and effectiveness [25], as follows. (1) Efficiency refers to the user’s ability to achieve goals quickly and accurately as well as the error rate during use. (2) Satisfaction relates to the user’s comfort and happiness, as well as meeting expectations and fulfilling needs. (3) Effectiveness refers to the ability to accomplish tasks within a certain time frame. The two reviewers categorized the research results based on the aforementioned definitions of the components of usability. In cases of any inconsistencies in the classification, a third reviewer reassessed the cases until a consensus was reached. This systematic approach ensured a thorough and standardized evaluation of the usability of mHealth apps.

## 3. Results

### 3.1. Characteristics of the Included Studies

Eleven articles published between 2013 and 2023 were included in the final review. Table 1 presents the characteristics of each mHealth app. Three studies were conducted in Mexico [29,30,31], two in Canada [32,33], two in China [10,34], and one each in Korea [35], the United Kingdom [11], Thailand [7], and Italy [36] (Table 1). The study designs varied, with three using experimental designs (randomized controlled trials (RCTs) [10,32,35], six using observational–analytic designs [7,29,31,33,34,36], and two being observational–descriptive studies [11,30]. mHealth apps were used on various devices: 27.2% (3/11) on smartphones, 18.2% (2/11) on tablets, 9.1% (1/11) on computers, and 72.7% (8/11) on unspecified devices. In terms of operating systems, 45.5% (5/11) were Android-based, 36.4% (4/11) were iOS-based, 36.4% (4/11) were Internet-based, and 45.5% (5/11) were unspecified. The measurement durations of these studies ranged from 10 weeks to 285 months. The interrater agreement was analyzed to assess interrater reliability among reviewers, and values ranged from 0.95 to 1, indicating excellent interrater reliability.

### 3.2. Interventions Description

Regarding the functionality of mHealth apps, each app included “inform”, “instruct”, and “communicate” features with a 100% inclusion rate (11/11). Regarding other features, “display” was in 90.9% (10/11) of the apps, “guide” in 90.9% (10/11), “record” in 81.8% (9/11), and “remind/alert”—the least common feature—in 72.7% (8/11) of the apps (Table 2). The intervention content of these apps included disease-related information. Patients could record and transmit details such as the fluid volume during dialysis or the condition of dialysis wounds. Additionally, in the case of symptoms or problems during dialysis, patients could communicate and interact with HCP or peers. The interrater agreement was analyzed to assess interrater reliability among reviewers, whose values ranged from 0.95 to 1, indicating excellent interrater reliability. 

### 3.3. Outcomes

The outcomes were categorized into three main types: efficiency, satisfaction, and effectiveness (Table 3). In one study that investigated PD-related self-efficacy, no statistically significant differences were found [35]. Two studies that explored usability drew conflicting conclusions, with one study reporting positive [30] and the other negative usability [7]. All five studies that assessed satisfaction reported positive outcomes [10,11,29,33,36]. For acceptance, which was studied in one study, positive results were reported [29]. Regarding effectiveness, a lesser number of patients from the intervention group switched from PD to HD or kidney transplantation compared with those from the control group [34]; however, three studies reported no significant differences [10,31,36]. In two studies, re-hospitalization rates showed no significant differences [10,31]. The peritonitis occurrence rates showed no significant differences in the two studies [31,36]. Infection rates at the exit site and mortality rates were examined in two studies, with one showing better outcomes in relation to fluid overload and inadequate solute clearance in the experimental group [34], and the other reporting no significant differences [10]. Inconsistent results for biochemical parameters were found across studies. Serum albumin and hemoglobin levels were higher in the experimental group in two studies [10,35]. However, changes in serum calcium levels did not differ significantly in two studies [10,32]. Mixed results were obtained for serum phosphorus levels, with two studies reporting no significant differences [32,35] and one showing better outcomes in the experimental group [10]. While no significant differences were found in the changes in serum potassium levels in a study [35], the calcium-phosphorus level was better in the experimental group in another study [10]. Taking calcium carbonate yielded no significant differences [32]. PD-related knowledge and health behaviors were better in the experimental group in one study [35]. Quality of life results varied, with one study showing better outcomes in the experimental group [35] and two studies reporting no significant differences [11,33]. No significant differences were observed for the consumer quality index [33] and technology readiness [32]. 

## 4. Discussion

In this study, a systematic review was conducted to identify and map the existing evidence regarding the features, functionalities, and usability of mHealth apps for patients with PD. The findings reveal that the apps primarily focused on self-efficacy in performing PD procedures, user satisfaction, and monitoring and improving health conditions. Regarding functionality, mHealth apps mostly encompassed features listed by the IMS. Regarding usability, the findings are similar to those found in Cartwright et al.’s (2021) systematic review on eHealth [8]; however, the results pertaining to the outcomes are inconsistent.

The most successful feature for mHealth apps should be multifunctionality [22]. In this systematic review, most apps were found to have multiple functions, such as record, display, guide, remind/alert, communicate [7,35], and facilitating remote monitoring [8]. They allow patients to upload data, such as fluid volumes during the PD process, enabling communication and interaction between HCP and patients to ensure treatment outcome. HCP can also track patient treatment results [10]. These results are consistent with those of Eberle et al. (2021) who found that apps improved laboratory data and facilitated health promotion [37]. Moreover, these apps provide disease-related information for patients with PD, resulting in increased user satisfaction.

The usability of an app can be assessed by exploring three aspects: efficiency, satisfaction, and effectiveness. Some results obtained in this review seem to differ from those in previous studies. In terms of efficiency, the apps did not seem to enhance patients’ self-efficacy. These results are inconsistent with the findings of Eberle et al. (2021), who indicated that patients with diabetes who used mHealth apps strengthened their self-efficacy [37]. This inconsistency may be attributed to the means of communicating or interacting with HCP, which are limited to phone calls, text messages, or monthly interviews, contributing to patients feeling less confident about dealing with disease-related situations. This could have influenced their lack of self-efficacy [35]. Furthermore, there was only one study on self-efficacy, and the intervention duration was only ten weeks; therefore, confirming whether apps can enhance self-efficacy is difficult. Similar findings were observed regarding usability. Lukkanalikitkul et al. (2022) believed that poor usability is due to unsuitable software development and limited Internet speeds, which do not meet user needs [7]. However, Olivares-Gandy et al. (2019), who focused on the app-development process, including finding shortcomings for improvement and allowing for timely adjustments to the app based on patient needs, found the usability to be effective [30].

All study findings reported patient satisfaction with app usage [10,11,29,33,36]. The primary advantage of mHealth apps is that they are not bound by time or space constraints, allowing users to access disease-related information, record personal data (e.g., blood pressure, weight, and PD fluid), and monitor their health status at any time. Patients instantly communicated with HCP on encountering any problems or difficulties. Patient satisfaction significantly improved if the app features were comprehensive, practical, user-friendly, and suitable for patients [29,30].

The effectiveness of apps in improving biochemical parameters revealed inconsistencies; while Eberle et al. (2021) showed some improvements in patients with diabetes, no statistically significant differences were found in others [37]. Continuous monitoring of nutritional management can help maintain normal hemoglobin and albumin levels in patients [35]. Learning fatigue, resulting in a low willingness to use apps, may occur when patients need to learn how to use apps correctly, which could be a contributing factor to the lack of improvement in physiological health indicators. In addition, the mHealth apps’ design not aligning with the needs of patients and being difficult to operate may contribute to increased frustration and discouragement among patients. It could lead to no ideally effective outcomes [32]. However, in a four-year study about the usage of a PD app, Xu et al. (2022) found improvements in all physiological indicators, for example cause-specific mortality and all-cause and cause-specific permanent transfer to hemodialysis, suggesting that the consistent and long-term use of apps to record daily life activities may yield effective outcomes [34].

App usage can enhance patients’ quality of life, particularly in areas where the symptoms and disease impact daily living [22,35]. However, no significant differences were observed in users’ physiological and psychological domains of quality of life. Kiberd et al. (2018) indicated that the lower use of apps might be due to a high rate of patient dropout [33]. Farfan-Ruiz et al. (2021) indicated that improvements in users’ technological readiness might be associated with gender, educational level, and race [32].

## 5. Conclusions

Although the functional features of mHealth apps for patients with PD are mostly complete, their effectiveness varies. This inconsistency may be due to the fact that only three RCT studies were included in this review, of which two study samples, both in the experimental and control groups, comprised fewer than 30 participants, which might have led to poor-quality evidence for intervention outcomes. Comparing the usability of apps between the experimental and control groups, some studies showed no significant differences in effectiveness, whereas others did. Health outcomes with mobile apps may be better than or equivalent to those of traditional healthcare. Future research should include more studies with larger sample sizes to explore and verify the long-term effectiveness of mHealth apps. In this systematic review, user satisfaction with apps was found to be generally high. Thus, using these apps in the care of patients with PD to promote the self-management of diseases and enhance health outcomes is recommended.

This study adhered to a systematic literature review approach by conducting a thorough and comprehensive search of the most common academic literature databases. Although the study findings suggest potential progress in the health of patients with PD as a result of using mHealth apps, this study has some limitations. First, out of the eleven studies, only three studies were RCTs. This may have resulted in weak evidence for the study outcomes. However, owing to the limited research on mHealth apps, researchers have considered various study designs to broaden the scope of the systematic review. Nevertheless, this approach may introduce selection bias, and the overall quality of the evidence may be low. Second, grey literature, non-full text articles, non-English publications, and conference reports in the field were excluded from this review. This omission may have led to valuable research in this field being excluded; therefore, the results should be interpreted cautiously. Third, this review only included studies published within the last 10 years, potentially overlooking valuable research published earlier. Finally, studies from different countries and regions were within the scope of the review, making it challenging to generalize the findings.

The ideal mHealth apps should have the functionality to record and monitor patients’ physiological conditions and provide reminders or alerts regarding abnormal situations to patients and HCP. More importantly, they must feature interactive functions with HCP to increase patients’ willingness to use them. Additionally, designing user-friendly software from the perspective of the patients is crucial. Only through user-friendly design can the mHealth apps fully function as an auxiliary tool for promoting healthcare.

## Figures and Tables

**Figure 1 healthcare-12-00593-f001:**
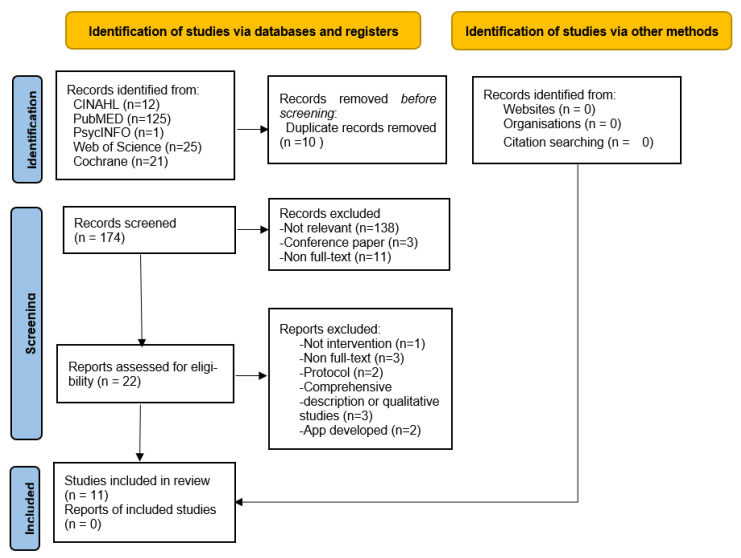
Flow diagram of study selection.

**Table 1 healthcare-12-00593-t001:** Characteristics of the included studies on patients with peritoneal dialysis (PD).

Authors/Year/Country/Ref No.	App Name/Device/Platform/Objectives	Design/Sample Size/Duration	Intervention Content	Outcome Variables/Results
Cao et al./2018/China [10]	The QQ Application Smartphone and tablet Android Investigate the effectiveness of the QQ application.	A randomized controlled trial design The QQ (experimental) group (n = 80) The traditional follow-up group (n = 80) 11.4 months	Nurses provide health education and disease-related information to patients through the QQ application, engaging in online conversations to address their health concerns.	Satisfaction, underwent kidney transplantation and switched to hemodialysis, rehospitalization, infection rates, serum values (1) Patients in the QQ intervention group showed significantly higher levels of serum albumin, hemoglobin, and satisfaction, as well as lower levels of phosphorus and calcium-phosphorus products compared to those in the control group. (2) There was no difference in underwent kidney transplantation and switched to hemodialysis, as well serum calcium levels between the two groups.
Chae and Kim/2023/Korea [35]	PD With You N/A Android 9.0 Developing a mobile application for enhancing self-management and assessing its effectiveness for patients with PD.	A randomized controlled trial The experimental group (n = 27) The control group (n = 26) 10 weeks	Patients used the self-management mobile application to record PD dialysis (i.e., replacement time and amount of removed fluid) and observe the physical indicators (i.e., body weight and blood pressure).	Self-efficacy, serum values, knowledge, health behavior, QOL (1) The experimental group showed a significant improvement in PD-related knowledge, PD-related health behavior, serum albumin, hemoglobin levels, and the domain symptoms/problems of kidney disease and disease impact on daily activity of HRQOL compared to the control group. (2) There were no significant differences between the two groups in PD-related self-efficacy, serum K, serum P, and the physical, psychological, and burden of kidney disease of HRQoL domains.
Dey et al./2016/UK [11]	N/A Computer tablets N/A (1) The application can provide information related to diseases and dietary considerations. (2) Patients can modify the dialysis plan or perform other actions, such as manual exchanges when dialysis resulted in problems. (3) Providing dietary advice through phone consultations by HCP. (4) Recording and accessing personal physiological data and details of dialysis sessions.	Observational–descriptive studies (a cross-sectional study) 22 participants Over 15 months	Each patient was provided with a weight scale, blood pressure monitor, and computer tablet. The patients were required to record data, such as weight, blood pressure, dialysis exchanges, and ultrafiltration volume. Additionally, patients had to report on the presence or absence of symptoms, such as swelling, shortness of breath, fever, abdominal pain, tenderness around the catheter site, and other relevant conditions. When the patients’ condition was not well, HCP provided interventions or treatments.	Satisfaction, QOL (1) No significant improvement in QOL. (2) No significant improvement in satisfaction with assistive technology.
Kiberd et al./2018/Canda [33]	McKesson eHealth portal applications N/A Investigate whether eHealth portal applications can effectively enhance the home dialysis care experience for patients.	Observational–analytic design, without a control group (a single-arm pilot study) Participants (n = 27) 12 months	Patients and HCP could communicate through a web portal application. HCP could provide suggestions such as medication changes, post-clinic visit explanations, new appointment scheduling, and more.	Satisfaction, acceptance, QOL, consumer quality index (1) Most participants were satisfied. (2) QOL did not improve, and it was difficult to ascertain the user acceptance due to the small sample size. (3) No significance in the consumer quality index.
Lukkanalikitkul et al./2022/Thailand [7]	CKD-PD app Smartphone and near-field communication and optical character recognition Android and iOS Identify chronic kidney disease—peritoneal dialysis application with near-field communication and optical character recognition functions, which can automatically gather the hydration status to enhance the care for patients with PD and improve it.	Observational–analytic design (user-centered design study) Participants (n = 10) 12 months	Provide an application to collect the peritoneal dialysis fluid and upload the data to HCP.	Usability In the end, participants showed decreased interest in using this app. Especially, entering the data into the NFC and OCR system of the app was difficult for participants compared to the manual data.
Martínez García et al./2018/Mexico [29]	N/A A mobile Web application (for HCP) and Android application (for patients) Android Evaluating the usability of a remote monitoring system for patients undergoing peritoneal dialysis treatment.	Quasi-experimental design (a case study) Participants (n = 24) 9 months	Providing applications to patients to monitor the patients’ condition and assess the usability of the app.	Satisfaction, acceptance (1) 94.5% of participants were satisfied with the app among patients with APD and 92.3% among CAPD. (2) 89.5% of participants accepted.
Olivares-Gandy et al./2019/Mexico [30]	N/A Android and iOS N/A Analyze, design, and develop a mobile health application for the dietary requirements of patients with PD, and then assess its usability and satisfaction.	Observational–descriptive (a case study) One patient and one nutritionist N/A	Patients’ experiences with the use of mobile applications.	Usability Patients found the application usable.
Polanco et al./2021/Mexico [31]	WhatsApp Computer, smartphone, tablet N/A The healthcare team utilized remote healthcare via WhatsApp to reduce hospital visits and the risk of infection during the COVID-19 pandemic.	Observational–analytic design (an observational prospective-longitudinal study) Participants (n = 946) 3 months	The patients sent daily dialysis records and photos of their lower limbs to HCP using WhatsApp. The medical team members utilized WhatsApp for communication and tracking the patients’ condition.	Switched to hemodialysis, rehospitalization, peritonitis rate The incidence rates of peritonitis, switched to hemodialysis, and hospitalization showed no difference.
Viglino et al./2020/Italy [36]	N/A N/A N/A Explore the reliability, safety, and effectiveness of Videodialysis assistance for PD patients; investigate the possibilities of the Videodialysis and whether it reduced the family care burden and recourse to nurses at home.	Observational–analytic design The intervention group (n = 15) The control group (n = 62) Follow-up at 285 months	The experimental group was provided with Videodialysis, a device consisting of two parts. One part included the equipment that patients should have at home, including a camera, monitor, microphone, and a technical connection box. The other part comprised the equipment required on the healthcare personnel’s end, including a high-resolution display, a network camera, a computer with speakers, and software capable of managing six audio streams simultaneously.	Satisfaction, switched to hemodialysis, peritonitis rate (1) All the patients expressed satisfaction with the app for enhance confidence. (2) There was no difference in the incidence of peritonitis. (3) Three out of the fifteen participants transitioned to hemodialysis.
Xu et al./2022/China [34]	Manburs N/A N/A Explore the long-term impact of telemedicine (Manburs app) on patients in terms of mortality and technical performance failure.	Observational–analytic design (a propensity-matched study) Participants (n = 7539) From June 2016 to December 2020 (4 years)	Telemedicine, through the application (Manburs), involves self-monitoring records, online educational materials, and real-time doctor–patient communication.	Switched to hemodialysis, infection rates, mortality, fluid overload, inadequate solute clearance The intervention group was observed to have significantly lower risks of all-cause mortality, CVD mortality, all-cause transfer to hemodialysis, transfer to hemodialysis from PD-related infection, severe fluid overload, inadequate solute clearance, and catheter-related noninfectious complications compared with the control group.

Notes: PD: peritoneal dialysis; N/A: not applicable; HCP: health care providers; QOL: quality of life; APD: automatic peritoneal dialysis; CAPD: continuously automatic peritoneal dialysis; CVD: cardiovascular diseases.

**Table 2 healthcare-12-00593-t002:** IMS Functionality of mHealth apps for patients with peritoneal dialysis.

Authors (Year)	Functionality						
	Inform	Instruct	Record	Display	Guide	Remind/Alert	Communicate
Cao et al. (2018) [10]	✓	✓	✓	✓	✓	✓	✓
Chae and Kim (2023) [35]	✓	✓	✓	✓	✓	✓	✓
Dey et al. (2016) [11]	✓	✓	✓	✓	✓	✓	✓
Farfan-Ruiz et al. (2021) [32]	✓	✓		✓	✓		✓
Kiberd et al. (2018) [33]	✓	✓	✓	✓	✓	✓	✓
Lukkanalikitkul et al. (2022) [7]	✓	✓	✓	✓	✓	✓	✓
Martínez García et al. (2018) [29]	✓	✓	✓	✓	✓	✓	✓
Olivares-Gandy et al. (2019) [30]	✓	✓	✓	✓	✓	✓	✓
Polanco et al. (2021) [31]	✓	✓	✓	✓	✓	✓	✓
Viglino et al. (2020) [36]	✓	✓		✓	✓		✓
Xu et al. (2022) [34]	✓	✓	✓				✓

**Table 3 healthcare-12-00593-t003:** Usability of mHealth apps for patients with peritoneal dialysis.

Outcomes		Statistical Significance		
		Positive	Negative	No Difference
Efficiency	PD-related self-efficacy			Chae and Kim (2023) [35]
	Usability	Olivares-Gandy et al. (2019) [30]	Lukkanalikitkul et al. (2022) [7]	
Satisfaction	Patients’ satisfaction	Cao et al. (2018) [10] Dey et al. (2016) [11] Kiberd et al. (2018) [33] Martínez García et al. (2018) [29] Viglino et al. (2020) [36]		
	Acceptance	Martínez García et al. (2018) [29]		Kiberd et al. (2018) [33]
Effectiveness	Underwent kidney transplantation and switched to hemodialysis	Xu et al. (2022) [34]		Cao et al. (2018) [10] Polanco et al. (2021) [31] Viglino et al. (2020) [36] ^b^
	Rehospitalization			Cao et al. (2018) [10] Polanco et al. (2021) [31]
	Peritonitis rate			Polanco et al. (2021) [31] Viglino et al. (2020) [36]
	The infection rates at the exit site	Xu et al. (2022) [34]		Cao et al. (2018) [10]
	Mortality	Xu et al. (2022) [34]		Cao et al. (2018) [10]
	Fluid overload	Xu et al. (2022) [34]		
	Inadequate solute clearance	Xu et al. (2022) [34]		
	Serum albumin	Cao et al. (2018) [10] Chae and Kim/2023 [35]		
	Serum Hemoglobin	Cao et al. (2018) [10] Chae and Kim (2023) [35]		
	Serum calcium			Cao et al. (2018) [10] Farfan-Ruiz et al. (2021) [32]
	Serum phosphorus	Cao et al. (2018) [10] Farfan-Ruiz et al. (2021) [32] ^a^		Chae and Kim (2023) [35]
	Serum potassium			Chae and Kim (2023) [35]
	Calcium-phosphorus product	Cao et al./2018 [10]		
	Taking calcium carbonate			Farfan-Ruiz et al. (2021) [32]
	PD-related knowledge	Chae and Kim (2023) [35]		
	PD-related health behavior	Chae and Kim (2023) [35]		
	Quality of life	Chae and Kim (2023) [35] ^a^		Dey et al. (2016) [11] Kiberd et al. (2018) [33]
	Consumer quality index (CQI)			Kiberd et al. (2018) [33]
	Technology readiness			Farfan-Ruiz et al. (2021) [32]

Notes: Positive: the intervention group was better than the control group; Negative: the control group was better than the intervention group; ^a^: mixed effects; ^b^: not clear.

## Data Availability

The data that support the findings of this study are available from the first author.

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
