# Peer review of "Functionality and Usability of mHealth Apps in Patients with Peritoneal Dialysis: A Systematic Review"

_healthcare, 2024, doi:10.3390/healthcare12050593_

Round 1

Reviewer 1 Report

Comments and Suggestions for Authors

The authors tackle a relevant research question and niche in mhealth research. The level of maturity of the field justifies a literature review. I do have two suggestions:

First, I have concerns regarding the contribution. This is not the first scoping review of mhealth and chronic dialysis https://mhealth.jmir.org/2020/4/e15549, nor of ehealth and peritoneal dialysis at home https://journals-sagepub-com.ezproxy.lib.torontomu.ca/doi/full/10.1177/0896860820918135. What is the position of this paper in relation to these previous attempts?

Second, in terms of methodology.: be mindful that a scoping review is not going to “determine the overall impact of mhealh apps”. This is not a meta-analysis. I am not sure that the authors clearly understand what a scoping review is. The two sources, which they cite, define scoping reviews as a mapping exercise, not of the outcomes, but of the research field. This is not a literature review method for informally aggregating outcomes and drawing inferences.

I suggest that the authors either revisit their study as a true scoping review, adopting a less stringent structure in measuring their results to enable a broader representation of the various research endeavors in the field, or change their methodology to a systematic review and adopt the norms of that type of review.

Author Response

Thank you for your advice.

Just as the review mentioned, this article (ttps://mhealth.jmir.org/2020/4/e15549) focuses on both blood and peritoneal dialysis. Researchers aim to explore peritoneal dialysis, which differs from hemodialysis. For instance, in peritoneal dialysis, it is necessary to record the inflow and outflow volumes of the dialysate. Due to the presence of a dialysis catheter in the abdomen, there is an emphasis on maintaining aseptic techniques during the administration of peritoneal dialysis to prevent infection.

The researchers aim to investigate, from the perspective of patients, what content they feel is necessary and design what may require improvement in the mHealth application.

This article cannot be searched due to permissions (https://journals-sagepub-com.ezproxy.lib.torontomu.ca/doi/full/10.1177/0896860820918135). Therefore, we cannot read it.

The title has been revised in the manuscript.

Reviewer 2 Report

Comments and Suggestions for Authors

Thank you for your paper and I enjoyed reading it. I wish you every success as you continue working in this area.

The Background set out the context for this review, and the importance of mHealth for this population is clear. As you refer to both mHealth and eHealth as separate concepts, it would be useful to include a sentence clearly outlining the difference between the two.

There was some repetition between the search strategy (2.1) and the inclusion and exclusion criteria (2.2), so perhaps remove some of the duplication.

The PRISMA flowchart needs to be amended as 22 – 10 = 12, and 1+3+2+3+2 = 11, so the cell should read 11 articles excluded.

The review process was clear, systematic and rigorous. Well done.

Table 1: I suggest that the table is reformatted. As it is, it is difficult to determine where the outcomes of one study end and the other begins. I also suggest that the outcomes column is summarised in bullet format (similar to the Design/ sample size/ Duration column), so that it is a lot shorter. This should improve readability. If there is a still a discrepancy between the amount of content in each cell perhaps consider consolidating two columns into one, perhaps the objectives and the App name?

Table 3: As with table 1, please consider reformatting as words aligned left are easier to read in a table. Table 2 includes symbols, and so these are appropriate for aligning to the centre.

Line 293: clarify what you mean by the lower use of apps. The Kiberd study was a pilot, so was there a high attrition rate, lack of engagement, use of other modalities in preference to apps?

In the discussion section, I think it would be useful to identify, where possible, the specific features that might have explained a better outcome in one study over another. You did this in lines 277 – 278, where you summarised the characteristics of apps that proved more satisfactory. In lines 280 onwards, and 301 and 302 you state that some studies showed a change in effectiveness, whereas others didn’t. Why was this? Did the ‘effective’ studies do anything different to the ‘ineffective’ studies? Were the sample demographics similar? Were the features of the app similar?

I suggest you add a brief summary of recommendations, based on the findings from your review. This would be very helpful for anyone thinking of designing an mHealth intervention.

Minor editing issues:

Line 73, 74: insert ‘the’ - and the HCP

Line 76: insert ‘s’ – patients’

Line 96: singular – app’s

Line 161: singular – app’s functionality was, or: the apps’ functionalities were

Table 1: Cao et al – A randomized controlled trial design.

Line 251: close the brackets – (facilitate remote monitoring)

Author Response

Thank you for your advice. The comments Q1~Q7 have been revised.

Reviewer 3 Report

Comments and Suggestions for Authors

Thank you for the opportunity to review this manuscript. The systematic approach to reviewing PD mHealth apps focused on usability outcomes and functionality is very interesting. Overall this manuscript is well written  and the findings are of interest to the community. The only feedback I have is very minor:

1. Line 19 acronym PD used but not introduced.

2. From lines 22-30 if word count would allow, please provide n values.

3. Line 37 is there a reference to support claim regarding cost?

4. Is there any available research for Patient education focused on PD? Would be interesting to add to introduction.

5. Line 57, a little confused here. Should this be "PD mHealth delivers established health services"?

6. Lines 70-74 seems to be repeating what was already stated in previous paragraph.

For final editing, suggest tables to be in landscape format.

Comments on the Quality of English Language

Overall manuscript is very well written.

Author Response

Thank you for your advice.

The comments has been revised.
